# A Potential Role of Interleukin-5 in the Pathogenesis and Progression of Amyotrophic Lateral Sclerosis: A New Molecular Perspective

**DOI:** 10.3390/ijms25073782

**Published:** 2024-03-28

**Authors:** Anca Moțățăianu, Sebastian Andone, Adina Stoian, Rodica Bălașa, Adina Huțanu, Emanuela Sărmășan

**Affiliations:** 11st Neurology Clinic, Mures County Clinical Emergency Hospital, 540136 Targu Mures, Romania; anca.motataianu@umfst.ro (A.M.); sebastian.andone@umfst.ro (S.A.); adina.stoian@umfst.ro (A.S.); rodica.balasa@umfst.ro (R.B.); sarmasan.emanuela@gmail.com (E.S.); 2Department of Neurology, George Emil Palade University of Medicine, Pharmacy, Science and Technology of Targu Mures, 540142 Targu Mures, Romania; 3Department of Pathophysiology, George Emil Palade University of Medicine, Pharmacy, Science and Technology of Targu Mures, 540142 Targu Mures, Romania; 4Department of Laboratory Medicine, Mures County Clinical Emergency Hospital, 540136 Targu Mures, Romania; 5Department of Laboratory Medicine, George Emil Palade University of Medicine, Pharmacy, Science and Technology of Targu Mures, 540142 Targu Mures, Romania

**Keywords:** ALS, cytokines, IL-5, ALS progression rate, ALS pattern of progression

## Abstract

Cumulative data suggest that neuroinflammation plays a prominent role in amyotrophic lateral sclerosis (ALS) pathogenesis. The purpose of this work was to assess if patients with ALS present a specific peripheral cytokine profile and if it correlates with neurological disability assessed by ALSFRS-R, the rate of disease progression, and the pattern of disease progression (horizontal spreading [HSP] versus vertical spreading [VSP]). We determined the levels of 15 cytokines in the blood of 59 patients with ALS and 40 controls. We identified a positive correlation between levels of pro-inflammatory cytokines (interleukin [IL]-17F, IL-33, IL-31) and the age of ALS patients, as well as a positive correlation between IL-12p/70 and survival from ALS onset and ALS diagnosis. Additionally, there was a positive correlation between the ALSFRS-R score in the upper limb and respiratory domain and IL-5 levels. In our ALS cohort, the spreading pattern was 42% horizontal and 58% vertical, with patients with VSP showing a faster rate of ALS progression. Furthermore, we identified a negative correlation between IL-5 levels and the rate of disease progression, as well as a positive correlation between IL-5 and HSP of ALS. To the best of our knowledge, this is the first study reporting a “protective” role of IL-5 in ALS.

## 1. Introduction

Amyotrophic lateral sclerosis (ALS) is a complex neurodegenerative disease that primarily impacts motor neurons (MNs) in the motor cortex, brainstem, and spinal cord. The pathophysiological mechanisms of MN degeneration in ALS are intricate, with neuroinflammation being a hallmark that involves the activation of immune cells like microglia, T cells, and astrocytes in the central nervous system (CNS), while other contributing factors to neurodegeneration in ALS include abnormal protein aggregation, glutamate excitotoxicity, oxidative stress due to MNs’ metabolic activity and limited antioxidants, mitochondrial dysfunction, and ribonucleic acid (RNA) dysregulation [1,2,3].

The adaptive immune system’s role in ALS is gaining attention due to changes in inflammatory markers and the peripheral immune system contributing to MN death [4,5,6]. Naïve T CD4+ lymphocytes, primary regulators in the adaptive immune response, differentiate into subsets, like T helper (Th) 1, Th2, Th17, Th9, Th22, and regulatory T cells (Tregs) influenced by intracellular transcription factors and cytokines, the latter one with immunosuppressive functions and showing altered population in ALS, suggesting a role in modulating the immune response [7,8,9,10,11].

Recent research indicates that CD4+ and CD8+ T cells can infiltrate the CNS in neurodegenerative diseases; activated T cells stimulate microglia, astrocytes, and macrophages through cell surface ligands and cytokines [12,13,14,15]. In ALS, immune and glial cells produce various cytokines, and pro-inflammatory ones, like interleukin (IL)-1β, IL-2, IL-8, IL-17, IL-33, and tumour necrosis factor-alpha (TNF-α), contribute to neuroinflammation, while anti-inflammatory cytokines, such as IL-4, IL-5, IL-10, IL-11, IL-13, IL-35, and transforming growth factor-β (TGF-β), exert protective effects [4,16,17,18,19].

Mounting evidence in ALS suggests that immune cells (Th1, Th2, and Th17) and cytokines have a dual role, displaying both protective and detrimental effects on MNs and CNS cells [18,20]. Th1 may contribute to neuroinflammation, releasing pro-inflammatory cytokines like interferon gamma (IFN-γ) and TNF-α, thus potentially worsening neuronal degeneration. In contrast, Th2 cells, producing cytokines, such as IL-4, IL-5, and IL-13, may have neuroprotective effects, thus counterbalancing pro-inflammatory effects and promoting an anti-inflammatory environment. Maintaining a Th1/Th2 balance is crucial for immune homeostasis [21,22,23].

Microglia in ALS exhibits a dual role, manifesting two functional states. M2 microglia are characterized by anti-inflammatory and protective functions and release IL-10 and TGF-β. Conversely, M1 microglia, with pro-inflammatory characteristics, release TNF-α, IL-1β, IL-6, and nitric oxide (NO). The “M1/M2 polarization” of microglia in ALS is dynamic and can undergo changes over time, thus influencing the disease course [24,25,26,27,28].

ALS is a complex and multifactorial disease, and the exploration of various immune responses, including the interaction between Th1 and Th2 phenotypes, is ongoing.

A deepening of our understanding of the neuroinflammatory process in ALS, viewed through the prism of cytokines and their impact on patients’ clinical status, could illuminate the pathophysiological mechanisms underlying the condition, thereby broadening perspectives on future patient management. By analyzing various parameters to assess patients’ clinical status and correlating them with serum levels of various cytokines, we aimed to elucidate a portion of the enigma surrounding the myriad processes involved in the onset, phenotype, and progression of ALS.

## 2. Results

### 2.1. Demographic and Clinical Data

#### 2.1.1. Participant Characteristics

Clinical characteristics and demographic data of the study population (59 ALS patients and 40 controls) are presented in Table 1. There were no significant differences in age and sex between ALS patients and controls. In the case group, 63% were men (37 out of 59 patients), resulting in a female/male ratio of 0.59. The mean age of ALS patients was 57 ± 8 years, with an age at disease onset of 55 ± 9 years and an age at ALS diagnosis of 57 ± 10 years.

The median survival time from the onset of the disease was 42 months, with a mean survival time of 49 months. From the diagnosis of the disease, the median survival time was 30 months, with a mean survival time of 35 months.

#### 2.1.2. Cinical Assessments

The mean ALSFRS-R was 38 ± 6 points. Among the ALS patients, 81% (48 of 59) were classified as spinal onset, and 19% (11 of 59) as bulbar onset. Regarding the ALS phenotype, 61% of patients had the classical phenotype, 34% had LMN-predominant involvement (18% with a flail arm phenotype, 16% with a flail leg phenotype), and 5% had a bulbar phenotype.

The BDI scores ranged from a minimum of 0 points to a maximum of 35 points, with a median value of 12 points. For the FAB, scores ranged from a minimum of 10 points to a maximum of 17 points, with a median score of 14 points.

The ΔPR ranged from a minimum of 0.08 to a maximum of 4.5, with a median value of 0.58. Notably, 40% of patients exhibited a disease progression rate below 0.47, 27% had a progression rate between 0.47 and 1.11 (inclusive), and 33% had a progression rate above 1.11.

In our ALS cohort, the spreading pattern was as follows: HSP in 42% (16% with cervical to contralateral cervical progression, and 26% with lumbar to contralateral lumbar progression), and VSP in 58% (5% with cervical to lumbar progression, 17% with lumbar to cervical progression, 19% with bulbar to cervical progression, 2% with bulbar–lumbar progression, 10% with cervical–bulbar progression, and 5% with lumbar–bulbar progression) (Figure 1 and Figure 2).

### 2.2. Correlation of Cytokine Profile with Clinical and Demographical Characteristics

While our findings yielded only moderate and weak results, they bear statistical significance. We endeavor to present them for two principal reasons: firstly, due to the limited number of articles within the specialized literature that explore the variations of pro-inflammatory and anti-inflammatory cytokines alongside clinical and demographic parameters in ALS patients. Secondly, we aim to elucidate the limitations we encountered, thereby providing valuable insights for future researchers. By addressing these limitations, future investigations may aim for greater precision in replicating our findings. Hence, despite the modest statistical outcomes, we shall delineate the correlations unearthed in this study. These findings hold potential to illuminate new avenues in the extensive pathophysiology of ALS.

We analyzed the cytokine profile in ALS, including pro-inflammatory cytokines (IL-1β, IL-2, IL-6, IL-17E/IL-25, IL-17F, IL-31, IL-33, GM-CSF, TNF-α), anti-inflammatory cytokines (IL-4, IL-5, IL-10, IL-13), cytokines assisting in TCD4+ modulation and programming (IL-12), and the cytokines characterizing specific TCD4+ cell responses (IFN-γ).

In the ALS group, the levels of IL-12p70 (*p* = 0.007) and TNF-α (*p* = 0.022) were significantly higher compared to the control group. Conversely, in the control group, the levels of IL-13 (*p* = 0.01) and IL-4 (*p* = 0.047) were significantly higher compared to the ALS group. No other cytokine showed statistically significant differences between the two groups.

In the ALS patients group, we assessed correlations between the serum levels of cytokines and various clinical parameters.

Correlation analysis between cytokines and demographic data revealed a positive correlation between the age at disease onset and IL-17F (r = 0.32, *p* = 0.01, CI (95%) = [0.06; 0.54]), IL-33 (r = 0.32, *p* = 0.01, CI (95%) = [0.06, 0.53]), IL-17E/IL-25 (r = 0.31, *p* = 0.01, CI (95%) = [0.05, 0.53]), and IL-31 (r = 0.26, *p* = 0.04, CI (95%) = [0.001; 0.49]). Additionally, there was a positive correlation between the age at ALS diagnosis and IL-17F (r = 0.34, *p* = 0.006, CI (95%) = [0.09; 0.56]), IL-33 (r = 0.33, *p* = 0.009, CI (95%) = [0.07, 0.54]), IL-17E/IL-25 (r = 0.35, *p* = 0.005, CI (95%) = [0.09, 0.56]), and IL-31 (r = 0.28, *p* = 0.02, CI (95%) = [0.02; 0.50]).

We observed a positive correlation between ALSFRS-R and IL-5 (r = 0.42, *p* = 0.0008, CI (95%) = [0.17, 0.61]). Additionally, a positive correlation was identified between ALSFRS-R-UL and IL-5 (r = 0.30, *p* = 0.01, CI (95%) = [0.04; 0.52]), along with a positive correlation between ALSFRS-R-R and IL-5 (r = 0.30, *p* = 0.02, CI (95%) = [0.03, 0.52]) (Figure 3). Furthermore, King’s staging showed a negative correlation with IL-10 (r = −0.26, *p* = 0.05, CI (95%) = [−0.50, 0.01]).

### 2.3. Correlation between Cytokine Profile concerning ALS Progression Rate and Survival Time

The ΔPR was negatively correlated with IL-5 (r = −0.26, *p* = 0.05, CI (95%) = [−0.50; 0.009]) (Figure 4).

We observed a positive correlation between survival from disease onset and IL-12p70 (r = 0.27, *p* = 0.04, CI (95%) = [0.002, 0.51]) (Figure 5a). Similarly, there was a positive correlation between survival from disease diagnosis and IL-12p70 (r = 0.32, *p* = 0.01, CI (95%) = [0.05; 0.54]) (Figure 5b).

A positive correlation was found between the progression pattern and IL-5, with patients with a horizontal pattern having a higher serum value of IL-5 (40.00 ± 64.15 pg/mL) compared to the vertical pattern values of IL-5 (16.88 ± 11.04 pg/mL) (*p* = 0.049) (Figure 6).

No statistically significant correlations were found regarding the time from disease onset to diagnosis, ALSFRS-R-B, ALSFRS-R-LL, BDI, FAB, and disease subtype.

## 3. Discussion

Ongoing research investigates the interplay between inflammation and the immune response to gain a better understanding of these mechanisms. It remains unclear whether inflammation is the primary trigger or a secondary response to neuronal degeneration, and current knowledge indicates a complex relationship between these factors [29,30]. Some researchers propose that the immune response and inflammation may act as primary triggers in ALS pathogenesis and progression; as such, abnormalities in the immune system, including the activation of immune cells and increased production of inflammatory cytokines, could contribute to the degeneration and death of MNs [31,32].

Recognition of neuroinflammation’s potential role in ALS has led to numerous studies revealing a connection between inflammatory factors and MN loss, even in the pre-symptomatic phase, as well as its association with disease severity [18]. Growing evidence suggests extensive neuroinflammation in ALS extending beyond the CNS to create a persistent pro-inflammatory environment in both peripheral and CNS tissues, thus contributing to the observed heterogeneity in ALS [33]. Research involving ALS patients and murine models has identified activated microglia and astroglia in the CNS, along with pro-inflammatory peripheral lymphocytes and macrophages, indicating a dual role for inflammation featuring an initial protective anti-inflammatory response followed by a subsequent toxic pro-inflammatory response [6,34].

The regulation of inflammation is dependent on the ability of microglia and macrophages to demonstrate distinct M1 and M2 phenotypes, which can, respectively, induce neurodegenerative or neuroprotective effects, the transition between M1 and M2 phenotypes being closely linked to neurodegenerative conditions, including ALS [25,35,36]. MNs release exosomes containing inflammatory molecules, which activate microglia and cross the blood–brain barrier. Activated peripheral macrophages can further infiltrate the CNS. Both types of immune cells release cytokines with either anti-inflammatory or pro-inflammatory properties, thereby influencing the resolution or exacerbation of neuroinflammation based on their functional roles and responses. Nevertheless, the precise temporal sequence and causal relationship between neuronal degeneration and neuroinflammation remain incompletely elucidated [37,38].

Recent studies on sporadic ALS patients have explored inflammatory and anti-inflammatory cytokine levels and their roles in disease progression. Elevated levels of inflammatory cytokines (IL-2, IL-6, TNF-α, and IFN-γ) in ALS patients have been found to correlate with disease severity, suggesting their potential as inflammation-related biomarkers [39,40]. Multiple clinical studies have shown that ALS patients exhibit increased peripheral immune response, with elevated levels of both pro-inflammatory (TNF-α, IL-1β, IL-2, IL-6, IL-7, IL-8, IL-12p70, IL-15, IL-18, IFN-γ) and anti-inflammatory cytokines (IL-4, IL-5, IL-10, IL-13) compared to healthy controls [18,41,42,43,44,45]. The presence of heightened levels of both pro-inflammatory and anti-inflammatory markers in ALS patients underscores their dual nature and the synergistic role of pro- and anti-inflammatory cytokines in the neuroinflammatory process.

In ALS-related neuroinflammation, an intriguing observation involves the behavior of IL-10 and IFN-γ. Despite IL-10’s typical anti-inflammatory function and IFN-γ’s usual pro-inflammatory role, both cytokines were found to be elevated in ALS patients. However, conflicting results have emerged from two distinct studies reporting lower serum levels of these cytokines in ALS patients [18,43]. Conversely, IL-6, a pro-inflammatory cytokine, tends to be elevated in ALS patients according to some studies [18,43,44], though not consistently across all research [46,47]. Additionally, ALS patients demonstrate lower serum levels of IL-33, a pro-inflammatory cytokine, as reported in one clinical study [48]. These findings highlight the intricate nature of neuroinflammation in ALS, underlining the need for comprehensive research into each contributing factor and its role in the immune process.

The presence of dual immune responses in ALS can be attributed to several factors: (1) the heterogeneity of ALS, (2) changes in disease progression that allow for the evolution of the immune response, (3) inflammation as a secondary response to MN degeneration, (4) genetic and environmental factors that influence the immune response, and (5) variability among ALS patients with different immune profiles due to genetic differences, disease duration, and other factors, such as patient age. Inflammation entails crosstalk where pro-inflammatory mediators can stimulate the production of anti-inflammatory cytokines and vice versa in an attempt to regulate the immune response [49].

Clinical studies investigating age-related variations in serum cytokine levels in ALS patients have not found significant differences for IL-1β, TNF-α, IL-10, IL-12, IL-23, IFN-γ, and IL-17A [18]. However, one study reported elevated serum levels of the pro-inflammatory cytokine IL-6 in older ALS patients, although these findings were not replicated in other research [46]. In our study, we identified a positive correlation between age of ALS onset and age at diagnosis of ALS and IL-17F, IL-33, IL-17E/IL-25, and IL-31, and we observed that older age at disease onset or diagnosis corresponded to higher serum values of these pro-inflammatory cytokines.

IL-17F induces expression of pro-inflammatory mediators in ALS and other neurodegenerative diseases, and elevated IL-17 levels are found in ALS patients [50,51,52,53,54]. IL-17, IL-31, and IL-33, as primary pro-inflammatory cytokines, indicate that older age at disease onset or diagnosis corresponds to a more pronounced pro-inflammatory response compared to younger patients. Research suggests that inflammation and immune system dysregulation may contribute to ALS progression, and the age of onset could influence the role of inflammatory cytokines in the disease process. Typically, older individuals may exhibit a higher degree of inflammation due to the aging process, commonly referred to as “inflammaging”, which could interact with the inflammatory aspects of ALS [55].

Several clinical studies of ALS patients have examined the link between functional status, assessed through ALSFRS-R, and serum cytokine levels, but only one study demonstrated a correlation between pro-inflammatory cytokines and disease severity as measured through ALSFRS-R and forced vital capacity [18]. Other studies have indicated that there were no significant associations between functional disability and IL-1β, IL-10, TNF-α, IL-6, IL-12, IL-17, IL-23, IL-15, and IFN-γ [18,46,54,56,57].

In our present study, we found a positive correlation between ALSFRS-R scores and IL-5 levels. Higher ALSFRS-R scores, indicating better clinical status, were associated with increased serum IL-5 levels. However, no correlations were observed with IL-1 beta, IL-10, TNF-α, IL-6, IL-12p/70, IL-17F, or IFN-γ. Moreover, weak but statistically significant positive correlations were noted between ALSFRS-R-UL and ALSFRS-R-R scores and IL-5, suggesting that higher scores in these categories, reflecting improved upper limb and respiratory function, are linked to elevated serum IL-5 levels.

Our research also revealed elevated IL-5 levels in ALS patients demonstrating a horizontal spreading pattern (HSP) of disease progression, with those exhibiting higher IL-5 levels experiencing prolonged survival. The presence of HSP was associated with increased survival, and among the cytokines examined, IL-5 consistently emerged as a predictor of favorable prognosis.

Although survival based on disease progression patterns has been extensively studied, there is limited literature available on the horizontal pattern (progression from one limb to the contralateral limb) and the vertical pattern (progression to an ipsilateral upper or lower limb) [58]. In a study involving patients with lower-limb-onset ALS, researchers found no statistically significant differences in survival between those with disease progression in the contralateral lower limb and those progressing in the ipsilateral upper limb [59].

Conversely, in another study involving ALS patients diagnosed with spinal onset, it was observed that patients with a vertical spreading pattern (VSP) have lower survival rates compared to those with an HSP [58]. Consequently, HSP is associated with better survival and slower disease progression.

Our research, for the first time, indicates that IL-5 plays a preferential role in determining the expression of an HSP, thus positively influencing disease progression and enhancing patient survival. Moreover, our findings regarding higher survival in patients with an HSP align with those reported in the cited study. The convergence of these IL-5-related data supports the assertion of an interconnection between this cytokine, the manifestation of an HSP, and the progression of the disease toward a slower course.

IL-5 has a specific impact on eosinophils and basophils within the hematopoietic lineage, distinguishing it from IL-3, IL-4, and GM-CSF [60,61]. IL-5 shares receptor components with IL-3 and GM-CSF, emphasizing a familial relationship, and it plays a role in preventing eosinophilic apoptosis during allergen-induced airway inflammation [62,63,64]. Predominantly secreted by CD4+ T cells, particularly Th2 cells, IL-5 is involved in the differentiation of CD4+ T lymphocytes. This differentiation, triggered by extracellular parasites or allergens, leads to the release of various effector cytokines [65,66].

In 2008, Beers et al. made a groundbreaking discovery in ALS using an experimental mouse model, thereby establishing the neuroprotective role of CD4+ T lymphocytes. They demonstrated that these cells suppress cytotoxic factors, thus extending disease duration and improving survival rates [67]. Therefore, an enhanced response of Th2 cells, derived from CD4+ T cells in reaction to extracellular parasites or allergens (or other unknown factors) and leading to increased IL-5 secretion, is associated with a more favorable clinical status in ALS patients. This novel theory not yet documented in the specialized literature aligns with researchers’ observations of the neuroprotective role of CD4+ T cells. It suggests that an inflammatory response akin to that triggered by allergens (though likely not caused by them) imparts a protective effect on the clinical progression of ALS and, consequently, on patient survival.

Research findings indicate the absence of the IL-5 receptor in microglia cells and the mitogenic effect of IL-5 on both macrophages and microglia [68].

In studies with cultured microglia from Wistar rats, researchers demonstrated that IL-5 induces an acceleration in microglial metabolism [69]. Despite the previously confirmed absence of IL-5 receptor gene expression in microglia, researchers demonstrated the lack of activation in the two signaling pathways associated with the IL-5 receptor (MAP kinases ERK1 and ERK2, as well as the Janus kinase–signal transducer and activator of transcription (JAK/STAT) 5A/5B) [69].

The JAK-STAT signaling pathway plays a crucial role in regulating genes during hormone release and inflammation in the CNS. Dysregulation of this pathway in the CNS is primarily associated with brain inflammation and the survival of neurons and glial cells [70,71]. IL-5 has been shown to activate the JAK-STAT pathway and inhibit ischemia-induced inflammation [72]. In Alzheimer’s disease, IL-5 demonstrates a protective function by reducing tau protein hyperphosphorylation and preventing cell apoptosis. The activation of the JAK2 pathway is crucial for IL-5’s neuroprotective effects in neurodegeneration [73]. However, the microglial receptor and intracellular signaling pathway essential for IL-5’s effects, including proliferation and increased microglial metabolism, remain unknown.

Further studies are needed to clarify the mechanism driving the preferential promotion of CD4+ T cell differentiation into Th2, leading to increased IL-5 secretion and contributing to microglial proliferation. Understanding the clinical implications of this mechanism, including its potential influence on disease progression patterns and the slower advancement of the condition, ultimately resulting in extended patient survival, is essential for unraveling the intricate pathophysiological cascade of this disease. Currently, our understanding of this complexity remains partial and limited.

In the specialized literature, only two clinical studies have reported results on the determining IL-5 serum levels to identify variations in this cytokine concerning the presence or absence of the specific disease. In one clinical study, serum levels of 10 cytokines were determined at a half-year interval. Initially, a statistically significant lower serum level of IL-5 was observed in ALS patients compared to the control group. However, at the second determination after 6 months, a lower level of IL-5 in ALS patients was found compared to the control group, but this difference was not statistically significant [74].

Contrastingly, in another study, the opposite phenomenon was observed compared to previous findings. A statistically significant higher serum level of IL-5, along with other inflammatory markers and cytokines, was identified in ALS patients compared to the control group [43].

Administering the humanized anti-IL-5 receptor alpha monoclonal antibody MEDI-563 reduced eosinophil levels in primates and showed a dose-dependent decrease in peripheral blood eosinophil in human with mild atopic asthma [75]. In a study using peripheral blood mononuclear cells, glatiramer acetate induced IL-5/IL-13 secretion, but treated patients exhibited a substantial reduction in these cytokines compared to healthy subjects. Glatiramer acetate administration increased serum IL-5/IL-13 levels in healthy subjects, untreated multiple sclerosis patients, and those with other neurological diseases, correlating with clinical efficacy [76].

A series of hypotheses have emerged from our research and have the potential to revolutionize ALS patient management, thus potentially leading to enhanced disease progression and increased patient survival. Allergenic stimulation correlates with a potential modulation of disease progression in ALS patients. Facilitating the differentiation of CD4+ lymphocytes toward Th2 cells and Tregs could represent a novel therapeutic target for ALS. IL-5 and its associated cytokine, IL-4, may play pivotal roles in decelerating disease progression in ALS patients. The intracellular signaling pathway of IL-5 in microglia could be activated to induce a neuroprotective response. An atopic predisposition might confer a protective effect against the progression of ALS. Besides allergens and parasites, other environmental factors may promote the differentiation of CD4+ T lymphocytes into Th2 lymphocytes, warranting investigation into their nature and influence. In light of the current study’s limitations, particularly its exclusive focus on demographic, clinical, and cytokine analyses of patients diagnosed with ALS, and given that the studies published in the specialized literature do not furnish us with answers concerning the hypotheses mentioned earlier, to address these queries, experimental studies should be conducted to either confirm or refute these hypotheses and shed light on the role of the mechanism in ALS pathophysiology.

However, it is imperative to prioritize the maintenance of immunological homeostasis among patients, particularly regarding allergic reactions. This equilibrium hinges on the delicate balance of Th2 and Treg populations, a phenomenon observed in both healthy individuals and those with allergies. This assertion is supported by a study encompassing cohorts of both healthy and atopic individuals [77].

While some limitations of this study were acknowledged beforehand, it is imperative to address others, as they may have impacted our results. One such limitation is the relatively small sample size, which could have influenced the accuracy of the statistical findings; notably, only moderate and weak correlations were observed among our results. Another limitation could stem from the wide age range of the participants, corroborated with the inflammaging phenomenon discussed earlier, thus potentially influencing cytokine levels. Therefore, multiple factors must be considered when interpreting the results, which may have hindered the attainment of robust findings. However, focusing on the most significant result obtained—the correlation between ALSFRS-R and IL-5—we can assert that, despite the limitations, we have identified a correlation between these two parameters for the first time. This discovery opens avenues for further research, with efforts aimed at addressing the aforementioned limitations to enhance the validity of future studies.

## 4. Materials and Methods

### 4.1. Participants and Assessment

This prospective, single-center study included 59 patients with sporadic ALS and 40 age- and sex-matched healthy controls. Conducted at the 1st Department of Neurology, Mures Clinical County Emergency Hospital, the study spanned from 2020 to 2022. All patients underwent clinical and electrophysiological (electromyography) evaluation by a single physician and investigator (AM). Inclusion criteria comprised the following: (1) signed informed consent, (2) age between 18 and 80 years, (3) diagnosis in our department with definite, probable (with probable laboratory support), or possible ALS, according to the El Escorial–Airlie House criteria [78], and a minimum of two clinical visits to our department at a minimum of 6-month intervals (needed to calculate the ALS Functional Rating Scale—Revised: ALSFRS-R). Exclusion criteria for both patients and the control group were as follows: (1) any clinical signs of acute infection in the past 6 weeks, (2) a personal history of other neurodegenerative, autoimmune, or inflammatory diseases, and (3) immune-suppressive therapy in the last 6 months (such considerations arise due to the potential impact on cytokine levels).

We recorded the following information for each patient during the study visit: age at the study visit (years), age at ALS diagnosis (years), gender, body mass index (BMI), disease subtype at onset (limb- or bulbar-onset ALS), disease duration at the study date in months (time from the first reported symptom to the date of the clinical evaluation), diagnostic latency in months (time from symptom onset to ALS diagnosis), site of the body region affected at onset, site of the body region involved in progression, bulbar symptoms, family history, medication, comorbidities, and ALS-related medical interventions, such as percutaneous endoscopic gastrostomy (PEG), non-invasive ventilation (NIV), or tracheostomy.

All patients underwent neurological evaluation, and the ALSFRS-R was calculated. The ALSFRS-R is a standardized assessment tool utilized for evaluating the functional status of individuals afflicted with amyotrophic lateral sclerosis (ALS). It comprehensively evaluates various functional parameters, including speech, salivation, swallowing, writing, manipulation of cutlery and kitchen utensils, dressing and personal hygiene, adjustment of body position in bed, arrangement of bedding, mobility, such as walking and ascending stairs, presence of dyspnea, orthopnea, and identification of respiratory failure. Based on the 12 items covering bulbar, upper limb, lower limb, and respiratory signs and symptoms, it is subdivided into 4 selective subscores: bulbar (ALSFR-R-B), lower limb (ALSFR-R-LL), upper limb (ALSFR-R-UL), and respiratory (ALSFR-R-R) [79]. The ALSFRS-R was recorded at the time of diagnosis and at the second study visit date. Subsequently, the ALSFRS-R progression rate (ΔPR) was calculated for the decline of ALSFRS-R from the ALS diagnosis to the date of sample collection, using the formula 48–[(ALSFRS-R at diagnosis − ALSFRS-R at study visit)/symptoms duration between symptom onset and study visit date (months)] [80]. To evaluate the differences between fast and slow ALS progression, we defined slow progression as ΔPR < 0.47 points/month, intermediate progression as ΔPR between 0.47 and 1.11 points/month, and fast progression as ΔPR > 1.11 points/month [81]. Each patient underwent psychological evaluation using the Frontal Assessment Battery (FAB) and Beck Depression Inventory (BDI).

Based on clinical and electromyographical findings during the initial consultation, ALS patients were divided into five phenotypes: (1) classical or typical, characterized by the onset of predominantly lower motor neuron (LMN) signs in two or more body regions with upper motor neuron (UMN) involvement in at least one region; (2) upper motor neuron (UMN) predominant involvement, but not exclusive; (3) lower motor neuron (LMN) predominant involvement, including flail arms and flail legs phenotypes; (4) bulbar phenotype, with onset and progression of bulbar signs, including dysphagia or dysarthria; and (5) ALS—frontal temporal degeneration phenotype (ALS-FTD) [82]. We applied the King’s College ALS clinical staging system to all patients, stratified into the following stages: stage 1 (symptom onset with involvement of the first region); stage 2A—diagnosis; stage 2B—involvement of the second region; stage 3—involvement of the third region; stage 4A—need for gastrostomy; stage 4B—need for non-invasive ventilation [83].

The spreading pattern of ALS was assessed by evaluating the extension of muscle atrophy or weakness beyond the site of onset, considering a combination of clinical, electrophysiological, and anamnestic data. ALS patients were classified into three groups based on the different directions of motor neuron degeneration spread from the onset: (1) horizontal spreading pattern (HSP)—cervical to the contralateral cervical region or lumbar to the contralateral lumbar region; (2) vertical spreading pattern (VSP)—cervical or lumbar region to the ipsilateral upper or lower limb, bulbar to lumbar/cervical region, or cervical/lumbar to the bulbar region; and (3) crossed spreading pattern (CSP) when clinical or electrophysiological signs spread in a diagonal progression model to the contralateral part [84].

Patients were stratified according to BMI into underweight (<18.5 kg/m^2^), normal weight (18.5–24.9 kg/m^2^), overweight (25.0–29.9 kg/m^2^), and obese (≥30 kg/m^2^) [85].

All requisite documentation was meticulously prepared, and the study received approval from the local ethics committees (Research Ethics Board of the Clinical County Emergency Hospital Mures and Research Ethics Board from the University of Medicine, Pharmacy, Sciences and Technology (UMFST) “George Emil Palade” Targu Mures). This research adhered to relevant guidelines and regulations, following the principles stated in the Declaration of Helsinki. The study protocol was meticulously elucidated to all participating patients, ensuring their comprehensive understanding before they provided their informed consent through signature.

### 4.2. Preparation of Biological Samples and Cytokine Assay

Venous blood samples (59 ALS and 40 controls) were collected in 7.5 mL clot accelerator vacutainers and centrifuged for 15 min at 2500× *g* at room temperature. The resulting serum was aliquoted and frozen at −80 °C for a maximum of 20 min after centrifugation.

The frozen serum samples were thawed 90 min before the cytokine measuring protocol was performed. The cytokine analysis was carried out at the Center for Advanced Medical–Pharmaceutical Research within the “George Emil Palade” UMFST Targu Mures using a multiplex immunoassay panel of 15 human cytokines, HCYTOMAG Millipore (IL-17F, granulocyte–macrophage colony stimulating factor (GM-CSF), IFN-γ, IL-10, IL-12p70, IL-13, IL-1β, IL-33, IL-2, IL-4, IL -5, IL-6, IL-17E/IL-25, IL-31, and TNF-α) (Merck KGaA, Darmstadt, Germany). The laboratory procedure was performed following the MILLIPLEX^®^ protocol, and the results were read on the Flexmap 3 D analyzer (Luminex^®^ xMAP^®^ technology, Luminex Corporation, Austin, TX, USA) with dedicated xPONENT^®^ software version 4.3.

### 4.3. Statistical Analysis

For the data analysis, we used specialised software tools like IBM SPSS Version 26.0 for Windows [86], and Microsoft Excel 2019 [87].

The data analysis encompassed determining frequency distributions, creating histograms and diagrams using Microsoft Excel, and establishing descriptive statistics (mean, median, standard deviation, and coefficient of variation).

We utilized the Spearman correlation coefficient (rho) to correlate quantitative variables, ensuring statistical significance with a *p*-value threshold of ≤0.05.

We used ANOVA tests to assess parametric variables while employing the same significance threshold. Descriptive statistics, including measures, such as mean, standard deviation, median, and min/max, were utilized to characterize the data’s distribution.

For categorical variables, we employed contingency tables and the Chi-square test.

Additionally, validity tests (ROUT test) and normality tests (Kolmogorov–Smirnov test) were conducted.

## 5. Conclusions

This study introduces a new molecular research approach to understanding ALS, suggesting a potential role for IL-5 and associated processes in maintaining clinical status. While hypotheses exist regarding connections between allergenic stimulation, immune modulation, and cytokine activity in ALS, current evidence supporting these links is inconclusive. Further research, including experimental studies and clinical trials, is needed to validate these hypotheses and gain a comprehensive understanding of their roles in ALS pathophysiology. Investigating the differentiation of CD4+ T cells into Th2, leading to increased IL-5 levels and microglial proliferation, could transform ALS patient management and enhance our understanding of the disease.

## Figures and Tables

**Figure 1 ijms-25-03782-f001:**
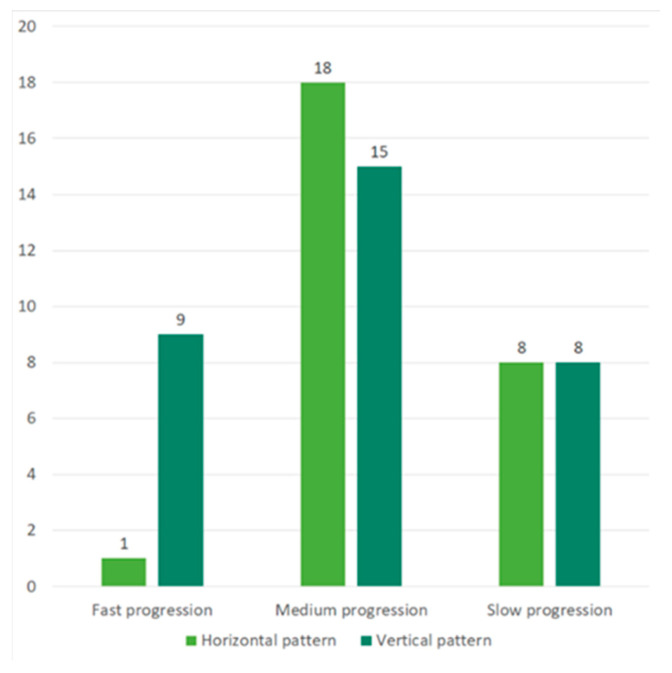
The horizontal and vertical spreading pattern subdivided into the 3 categories of the progression rate: fast, moderate, and slow (300 dpi).

**Figure 2 ijms-25-03782-f002:**
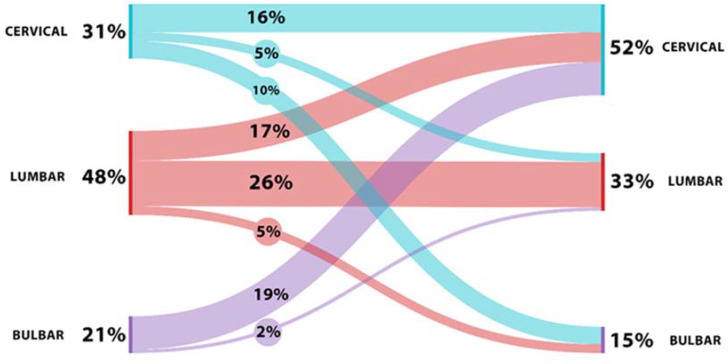
The spreading pattern of the disease based on the different directions of motor neuron degeneration spread from the onset and the percentage related to each variant of spreading (300 dpi).

**Figure 3 ijms-25-03782-f003:**
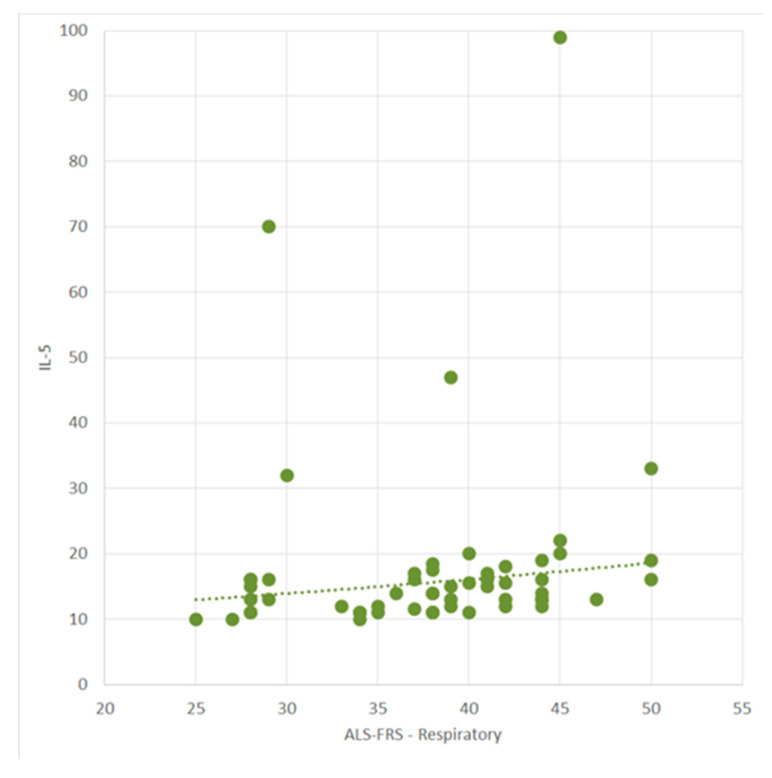
The correlation between respiratory ALSFRS-R and IL-5 (300 dpi).

**Figure 4 ijms-25-03782-f004:**
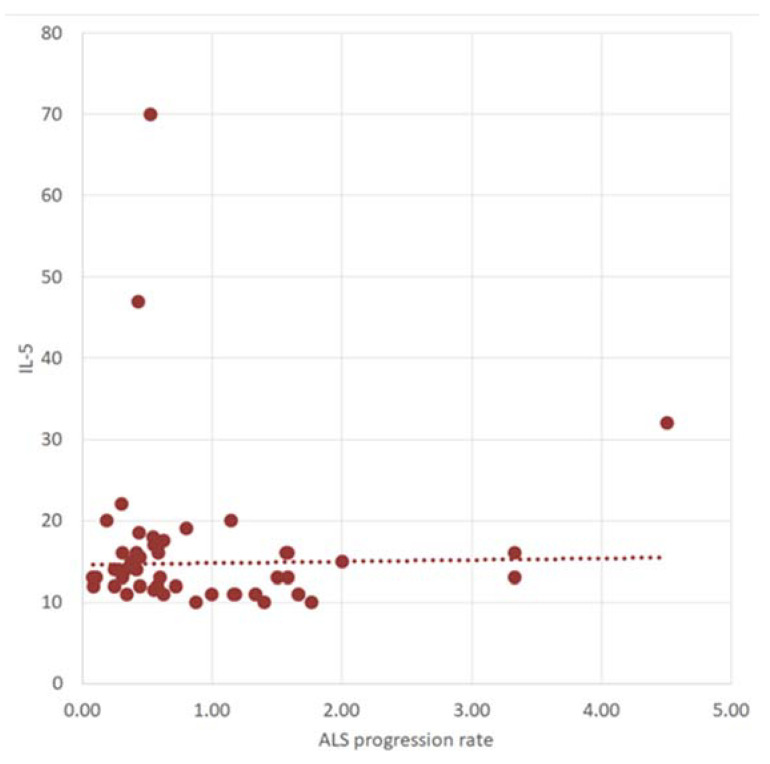
The correlation between ALS progression rate and IL-5 (300 dpi).

**Figure 5 ijms-25-03782-f005:**
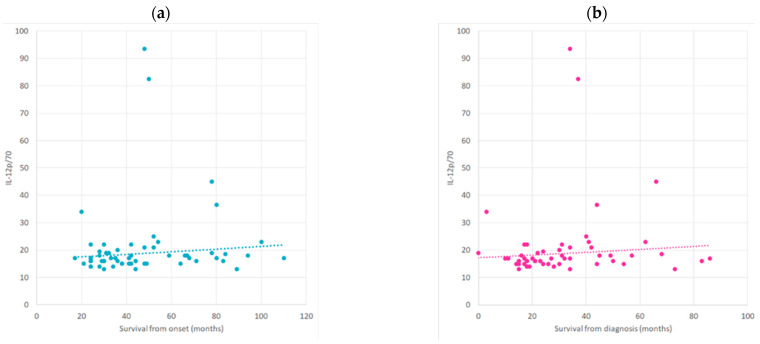
(**a**) The correlation between survival from the onset of the disease and IL-12p/70. (**b**) The correlation between survival from disease diagnosis and IL-12p/70 (300 dpi).

**Figure 6 ijms-25-03782-f006:**
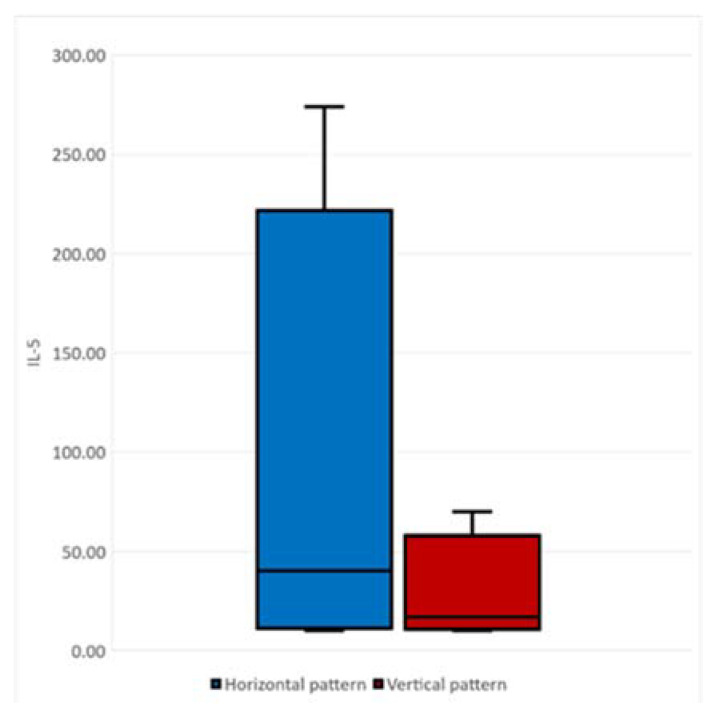
The correlation between the spreading pattern of the disease and IL-5 (300 dpi).

**Table 1 ijms-25-03782-t001:** Demographic and clinical data of enrolled cases.

Demographic and Clinical Data	ALS Cases	Controls
Number of patients n	59	40
Gender of patients n (%)		
Females	22 (37.3%)	18 (45.0%)
Males	37 (62.7%)	22 (55.0%)
Age years (mean ± standard deviation)	57.28 ± 9.79	56.45 ± 7.95
ALSFRS-R points (mean ± standard deviation)	38.05 ± 6.40	
Progression rate of ALSFRS-R		
Progression rate < 0.47%—Fast	40.0%	
Progression rate 0.47–1.11%—Moderate	27.3%	
Progression rate > 1.11%—Slow	32.7%	
ALS type n (%)		
Bulbar-onset	11 (18.6%)	
Spinal-onset	48 (81.4%)	
ALS phenotype %		
Flail arm	18.64%	
Flail leg	16.94%	
Bulbar	3.38%	
Typical (LMN and UMN involvement)	61.01%	
King’s staging %		
2A	8	
2B	46	
3	44	
4A	2	
Clinical progression pattern %		
HSP	42%	
Cervical to cervical contralateral	16%	
Lumbar to lumbar contralateral	26%	
VSP	58%	
Cervical–lumbar	5%	
Lumbar–cervical	17%	
Bulbar–cervical	19%	
Bulbar–lumbar	2%	
Cervical–bulbar	10%	
Lumbar–bulbar	5%	
Survival months		
From onset (mean ± standard deviation)	46.85 ± 22.93	
From diagnosis (mean ± standard deviation)	32.86 ± 19.69	
Beck’s Depression Inventory points (mean ± standard deviation)	15.53 ± 9.21	
Frontal Assessment Battery points (mean ± standard deviation)	13.73 ± 2.06 ^1^	

^1^ Abbreviations: n = number, ALS = amyotrophic lateral sclerosis, ALSFRS-R = ALS functional rating scale—revised, LMN = lower motor neuron, UMN = upper motor neuron, HSP = horizontal spreading pattern, VSP = vertical spreading pattern.

## Data Availability

The data presented in this study are available upon request from the corresponding author.

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
