# Peer review of "A Potential Role of Interleukin-5 in the Pathogenesis and Progression of Amyotrophic Lateral Sclerosis: A New Molecular Perspective"

_ijms, 2024, doi:10.3390/ijms25073782_

Round 1
Reviewer 1 Report
Comments and Suggestions for Authors
review report attched

suggested in review report
Reviewer 2 Report
Comments and Suggestions for Authors
This manuscript addresses an important and current issue about the role of immune regulation in ALS. I have concerns about the number of weak correlations presented. Several other papers have not yet found the significance of the circulating pro- and anti- inflammatory cytokines. Therefore, it would be important to make sure that there are stronger correlations between the cytokine profiles and ALS parameters.
Comments:
1. In the results, the demographic and clinical data was nicely presented. The Figure 1 was nicely presented and interesting.
2. The correlation analysis had many low r values suggesting many weak correlations (<0.3) to moderate correlations (0.5-0.3) but no strong correlations. The best positive correlation appeared to be ALSFRS-R and IL-5. The ALSFR-RUL, ALSFR-R-R, and ΔPR and IL5 all have weak correlations. Similar with several other cytokine correlations. Therefore, I am not sure there is strength in the association of IL-5 with ALSFR-R subscores or progression rate. Could the authors comment on this.
3. The last section of correlations between smoking, ETOH, and BMI also have weak correlations and could probably be omitted.
4. The discussion is very long and hard to follow. A more concise discussion surrounding the main positive findings of the paper would be more readable.
Round 2
Reviewer 1 Report
Comments and Suggestions for Authors
Article can be accepted for publication.
Author Response
We are grateful for your appreciation.
Reviewer 2 Report
Comments and Suggestions for Authors
Rereview:
1. 2. The correlation analysis had many low r values suggesting many weak correlations (<0.3) to moderate correlations (0.5-0.3) but no strong correlations. The best positive correlation appeared to be ALSFRS-R and IL-5. The ALSFR-RUL, ALSFR-R-R, and ΔPR and IL5 all have weak correlations. Similar with several other cytokine correlations. Therefore, I am not sure there is strength in the association of IL-5 with ALSFR-R subscores or progression rate. Could the authors comment on this.
Reviewer’s follow-up comment: The results should make some comment about the strength of the correlations. Authors could consider removing some of the very weak correlations with near flat-line graphs. Appreciate that the limitation of the study was added and included a statement about the weak correlations.
2. The discussion could do with shortening by a few paragraphs, concentrating on the relevance of the findings.
Comments on the Quality of English LanguageThere are minor typographical errors but overall English language is not a problem.
Round 3
Reviewer 2 Report
Comments and Suggestions for Authors
No new comments.